# Evaluation of Celligent^®^ Biomimetic Water Gradient Contact Lens Effects on Ocular Surface and Subjective Symptoms

**DOI:** 10.3390/diagnostics13071258

**Published:** 2023-03-27

**Authors:** Raúl Capote-Puente, José-María Sánchez-González, María Carmen Sánchez-González, María-José Bautista-Llamas

**Affiliations:** Optica Area, Vision Research Group (CIVIUS), Department of Physics of Condensed Matter, Faculty of Pharmacy, University of Seville, 41012 Seville, Spain

**Keywords:** pre-lens tear film, lipid pattern, non-invasive break-up time, contact lens

## Abstract

This study aimed to evaluate the non-invasive and subjective symptoms associated with Lehfilcon A water gradient silicone hydrogel contact lenses with bacterial and lipid resistance technology. A prospective, longitudinal, single-centre, self-controlled study was conducted among silicone hydrogel contact lens wearers. Non-invasive analysis of the pre-lens tear film was performed using the Integrated Clinical Platform (ICP) Ocular Surface Analyzer (OSA), and the meibomian glands were evaluated with the Cobra^®^ HD infrared meibographer. After 30 days of contact lens wear, the subjects were re-evaluated to determine the changes in conjunctival redness, subjective dry eye disease, tear meniscus height, lipid pattern, and non-invasive break-up time. Results showed that the lipid layer thickness decreased significantly from 2.05 ± 1.53 to 0.92 ± 1.09 Guillon patterns, and the tear meniscus height decreased from 0.21 ± 0.04 to 0.14 ± 0.03. The mean pre-lens non-invasive break-up time (NIBUT) significantly increased from 15.19 ± 9.54 to 25.31 ± 15.81 s. The standard Patient Evaluation of Eye Disease (SPEED) score also decreased from 7.39 ± 4.39 to 5.53 ± 4.83. The results suggest that Lehfilcon A significantly reduced lipid and aqueous tear film volume but improved break-up time and subjective dry eye symptoms.

## 1. Introduction

The success of soft contact lenses (SCLs) is largely determined by compatibility and how well they work with the ocular environment. It is influenced by multiple factors that do not disrupt the stability of the ocular surface [1,2]. Silicone hydrogel (SH) SCLs can interact with and modify biochemical tear film properties [3]. Prolonged and inappropriate use of these SH-SCLs could influence the ocular surface [4] and be associated with inflammatory variations [5], such as limbal and ciliary conjunctival redness [6], lacrimal morphology changes [7] or ocular surface disorders [8].

SCL modifies the precorneal lacrimal dynamics and divides it into one pre-lens and one post-lens, each with less thickness than the initial physiological thickness [9]. Concurrently, tear film proteins, lipids and carbohydrates that facilitate the adhesion, formation and maintenance of surface biofilms begin to be deposited on the SCL [10,11].

Specialised industries, including those in the field of contact lens manufacturing, are aimed at improving SCL properties that have the greatest impact on safety, comfort or tolerance of use, such as the module, contact angle, coefficient of friction and oxygen transmissibility [12,13]. However, current advances are targeted at materials and surface coatings that generate greater efficiency and safety on adhesion to lipids and biofilms that reduce the incidence of microbial keratitis or induced ocular redness [14,15]. Similarly, the inclusion of wetting agents and the incorporation of new dual structures, such as the aqueous gradient, have seemingly improved the hydrophobic value of the polymers, establishing it as a direct correlation on patterns, such as wettability [16,17], lubrication or dehydration of the SCL [18,19].

Tear film stability assessment with non-invasive automated technology is a potentially effective methodology to evaluate the ocular environment differences created by SCL [20]. The provided information on tear film kinetics and the resulting interpretations of tear film rupture and coverage cycles, or periods of atmospheric exposure between pre-SCL and precorneal flickers [21,22] can help to collect objective data on changes to lacrimal dynamics or the ocular surface, with or without previous symptoms [23,24]. Additionally, independently evaluating factors, such as evaporation rates, tear film volume, and meibomian expression, or using validated questionnaires, such as the Contact Lens Dry Eye Questionnaire (CLDEQ8) or Standard Patient Evaluation of Eye Dryness (SPEED) [25,26] can be used as a method to anticipate clinical management or to modify factors, such as adjustment, materials, mode of use, care systems, or even the discontinuation of SCL use [27,28]. The significance of the paper lies in evaluating the effectiveness of Lehfilcon A silicone hydrogel contact lenses, which have bacterial and lipid resistance technology, in reducing subjective symptoms of dry eye disease and improving non-invasive tear film analysis results. The motivation for conducting this study is to provide clinical evidence for the benefits of Lehfilcon A contact lenses in managing dry eye symptoms and maintaining ocular surface health.

The purpose of the study was to evaluate various parameters in individuals who wear silicone hydrogel contact lenses with Celligent^®^ biomimetic surface properties. These parameters included conjunctival redness classification, lipid layer thickness using an interferometer, tear meniscus height, the first and mean non-invasive break-up time (NIBUT), lid opening time, CLDEQ8, and SPEED. All these measurements were taken in order to gain a better understanding of the properties of these specific contact lenses and how they impact the eye.

## 2. Materials and Methods

### 2.1. Design

We conducted this prospective, longitudinal, single-centre self-control study with a timeline of 30 days. It was performed at the Optics and Optometry cabinets of the Pharmacy School (University of Seville). This research was conducted according to the Helsinki Declaration and the Ethical Committee Board of the University of Seville (0384-N-22).

### 2.2. Subjects

All the subjects included in the final analysis read and signed the informed consent form. An informative sheet was provided to all subjects with the detailed study procedure. The inclusion criteria were as follows: (1) healthy subjects without any eye disease or eye treatment, (2) age between 18 and 35 years old, (3) Contact Lens Dry Eye Questionnaire 8 (CLDEQ8) score above 0 points [29], (4) daily or monthly replacement silicone hydrogel contact lens wearers, (5) objective and subjective spherical equivalent refraction ≤5.50 diopters, and (6) objective and subjective refractive astigmatism ≤1.50 diopter. The exclusion criteria were as follows: (1) ocular infection, with no previous history of ocular surgery, (2) taking any ophthalmic or systemic medications with tear film or ocular surface effects, (3) Sjogren syndrome, (4) Rheumatoid arthritis, (5), diabetes, (6) Thyroid disorders, and (7) pregnancy or breastfeeding.

### 2.3. Materials

The non-invasive pre-lens tear film analysis was performed using the Integrated Clinical Platform (ICP) Ocular Surface Analyzer (OSA) from SBM System^®^ (Orbassano, Torino, Italy). The device was previously described in detail [30]. The meibomian gland evaluation was conducted using the non-mydriatic infrared meibography digital fundus camera Cobra^®^ HD (Construzione Strumenti Oftalmici CSO^®^, Firenze, Italy). The degree of meibomian gland dysfunction (MGD) was measured using the ImageJ method defined by Pult and Nichols [31]. MGD was classified into one of four grades according to the severity of the loss.

Two subjective dry eye disease questionnaires were used: the Contact Lens Dry Eye Questionnaire 8 (CLDEQ-8) [29] and the Standard Patient Evaluation of Eye Dryness (SPEED) [32,33] test. The silicone hydrogel contact lens used in the study was TOTAL 30^®^ (Alcon Inc., Fort Worth, TX, USA) with a monthly replacement schedule and daily wear. It belongs to the Food and Drug Administration (FDA) material group and has a high-water content, is non-ionic (V-B), and has Biomimetic Celligent^®^ Technology that provides resistance to bacteria and lipid deposits. The water gradient technology within the lens results in a high-water content (>90%) at the outermost surface.

All study subjects used the same multipurpose solution (MPS), Lens 55^®^ Care Hyaluropolimer Plus 360 mL, Servilens Fit and Cover^®^ (Granada, Spain), for contact lens care. The solution contained 0.00015% polyhexamethylene biguanide (PHMB), 0.01% ethylenediaminetetraacetic acid (EDTA), sodium hyaluronate, and hydroxyethyl cellulose in an isotonic, buffered aqueous solution.

### 2.4. Examination Procedure

In the first phase, subjects were included or excluded according to previously defined criteria. The subject’s samples were obtained from the non-optometry academic community. Standard contact lens protocol adaptation was performed according to the Graeme Young Soft Lens Design and Fitting chapter in the Nathan Efron Contact Lens Practice Book. All subjects were instructed to abstain from using any contact lenses, lubricants, or drops for a week prior to the study. After this wash-out period, they underwent a non-invasive examination and filled out subjective questionnaires, as previously described, to evaluate minor to major tear film fluctuations. The variables assessed were conjunctival redness classification (CRC), lipid layer thickness (LLT), tear meniscus height (TMH), first non-invasive break-up time (FNIBUT), mean NIBUT, lid opening time (LOT), CLDEQ-8, SPEED, and the degree of Meibomian gland dysfunction (MGD) measured using an infrared non-contact camera. Meibography was performed using a digital fundus camera (Cobra HD^®^).

In the second phase of the study, the subjects were re-examined after wearing contact lenses (Lehfilcon A) for 30 days. The aim was to evaluate the volume, stability, and lipid pattern of the short-term tear film in front of the lenses. All measures were performed with the contact lenses on, and the subjects were instructed to use only the provided lens care regimen and avoid using eye drops or lubricants. The temperature and humidity conditions were stable during all measurements, and the ocular surface test was alternated between the eyes. The subjects were asked to blink normally within one minute between OSA measurement steps and intentionally blink three times before the next measurement.

### 2.5. Statistical Analysis

Statistical analysis was performed using SPSS statistical software (version 26.0, IBM Corp, Armonk, NY, USA). Descriptive analyses were performed using the mean ± SD (range values). The normal distribution of the data was assessed using the Shapiro-Wilk test. Differences in qualitative variables were assessed using the chi-square test. Differences between previous, short-term, and one-month Pre-lens variables were performed using the Wilcoxon test. For all tests, the significance level was set at 95% (*p* value < 0.05). The sample size was estimated using the GRANMO^®^ calculator (Institut Municipal d’Investigació Mèdica, Barcelona, Spain. Version 7.12). A two-tailed test was used. Alpha and beta risks are set at 5% and 20%, respectively. The estimated standard deviation (SD) of the difference was set to 0.45 (based on the SD primary variable study by Marx et al. [34]), and the expected minimum NIBUT difference in front of the camera was set to 0.30 s, followed by the loss at the end. The upward adjustment rate has been fixed at 0.10. This resulted in a recommended sample size of 20 subjects.

## 3. Results

Thirty-one silicone hydrogel contact fittings were performed in a sample of thirty-one myopic with low astigmatism subjects (random eye selected). Descriptive analyses of sex, nationality, age, non-cycloplegic manifest refraction, LogMAR and decimal visual acuity, corneal meridian, contact lens power, and superior and inferior eyelid meibomian gland dysfunction are presented in Table 1. Longitudinal ocular surface measurements were performed before and after Lehfilcon A contact lens fitting and are presented in Table 2. All patients had well-fitting contact lens wear during the 30 days. One patient had a mild punctate erosion that resolved after 3 days of contact lens removal. Lens movement was correct in all cases. Subjects wore the contact lens for 5.61 ± 1.75 days per week, 8.95 ± 3.25 h per day and 3.68 ± 2.42 h on the one-month visitor day. A comparison between low wear time (under 8 h) and high wear time (above 8 h) was performed, but non-statistically significant differences were achieved between the follow-up variables.

### 3.1. Ocular Surface Analyzer

Conjunctival redness classification achieved a non-statistically significant increase of 0.13 ± 0.70 grades on the Efron Scale (W = 196.00, *p* = 0.14). A modest effect size of 0.22 was reported. Lipid layer thickness interferometry decreased 1.20 ± 1.39 grades on the Guillon Scale (W = 42.00, *p* < 0.01). A large effect size of 0.85 was reported. Lipid layer interferometry before and one month after CL is presented in Figure 1. The tear meniscus height significantly decreased with a change of 0.06 ± 0.05 mm (W = 71.50, *p* < 0.01). A large effect size of 1.97 was reported. TMH before and one month after CL is presented in Figure 2. FNIBUT described a small decrease of 0.31 ± 1.57 s (W = 591.00, *p* = 0.09). A 0.29 modest effect size was reported. MNIBUT achieved a statistically and clinically significant difference of 10.00 ± 15.93 s. (W = 1443.00, *p* < 0.01). A large effect size of 0.77 was reported. Finally, LOT showed the largest increase of 20.03 ± 26.50 s. (W = 1537.00, *p* < 0.01). A large effect size of 0.82 was reported. TMH, FNIBUT, MNIBUT and LOT longitudinal changes are presented in a box and plot graph in Figure 3.

### 3.2. Subjective Questionnaires

CLDEQ8 remained similar, with a slight decrease of 0.70 ± 9.24 score points (W = 767.00, *p* = 0.49). A trivial size effect of 0.11 was reported. Finally, SPEED significantly decreased within 1.83 ± 6.24 score points (W = 453.00, *p* <0.01). A modest size effect of 0.40 was reported. CLDEQ8 and SPEED longitudinal changes are presented in a box and plot graph in Figure 3.

## 4. Discussion

In this study, we assessed conjunctival redness, lipid layer thickness, tear meniscus height, first and mean NIBUT, lid opening time, CLDEQ-8 and SPEED in wearers of silicone hydrogel contact lenses with Celligent^®^ biomimetic surface properties and water gradient technology. Results showed a decrease in lipid layer thickness and tear meniscus height and a significant increase in mean pre-lens NIBUT. Scores on both the CLDEQ-8 and SPEED questionnaires also decreased after 30 days of wearing Lehfilcon A lenses.

Conjunctival redness is an indicator of a potential vascular response to various factors, such as the degree of oxygen transmission, which can lead to corneal hypoxia or allergic reactions and wettability issues [12,35]. Mechanical irritation caused by SH-SCL can trigger an adaptive immune response resulting in serious eye complications and discontinuation of SCL use [36]. The ocular surface is highly sensitive to mechanical irritation, which can induce an innate immune response [37]. This response triggers the production and release of pro-inflammatory cytokines, such as IL-1 and TNF-alpha. These cytokines promote the generation of pro-lymphangiogenic factors, which create channels for dendritic cells (DCs) to migrate to lymph nodes [38]. Furthermore, the pro-inflammatory cytokines can activate resident DCs, which then utilise afferent lymphatic vessels to travel to regional lymph nodes. This migration of DCs initiates the generation of effector CD4 T cells, including Th1 and Th17 cells, which are critical components of the adaptive immune response [39].

It is important to note that this cascade of events highlights the complex interplay between innate and adaptive immunity in response to ocular surface irritation. By understanding the underlying mechanisms, we can develop novel approaches to prevent and treat ocular surface diseases associated with inflammation and immune dysregulation [37,38].

Although elapsed SH-SCL use time is a determining factor in complication risks, some scientific trials confirmed increased bulbar redness after six hours of use [40,41,42]. In the current study, non-statistically significant changes in bulbar redness were observed after 30 days of use in 8 eyes and decreased in 16 eyes, and no changes were observed in 38 remaining eyes, which could be attributed to the last generation SH-SCL high oxygen permeability, as other researchers previously achieved [43,44,45].

The LLT was significantly lower with time and the use of Lehfilcon A, which would justify why the FNIBUT was lower. Further studies should be performed to prove that the biomimetic Celligent^®^ Technology treatment also allows for lipid and bacterial adhesion resistance, which can influence the decrease in LLT and, as a consequence, FNIBUT [46]. Mann et al. observed that when inserting CL, its properties, the individual characteristics of the patient and the program of use produce variations both in the ocular surface and in the tear film due to different biophysical changes [47].

The LOT showed significant increases in opening times between blinks in subjects carrying Lehfilcon A compared to the baseline results (W = 1537.00, *p* < 0.01), and there was simultaneously a greater atmospheric exposure to the desiccation of the anterior surface of the SH-SCL. Eftimov et al. [16] referred to the increases and statistical superiority produced mainly by the material (Delefilcon A) in the dehydration-rehydration cycles combined with other latest generation SH-SCL used in the in vitro test.

TMH decreases significantly; to keep the CL hydrated, it is necessary to use the tear film reserve of the TMH [48,49]. The stability of this prelental tear film improves user comfort by reducing symptoms, especially prior to the adaptation of CL [50].

The MNIBUT increases after 30 days of contact lens use. At this point, we agree with other authors, such as Vidal-Rohr et al. [12] and Müller et al. [51], regarding materials and surface treatments similar to the one studied in this work. Although they measured it after a few hours of use, Fujimoto et al. [19] reported results after 30 ± 5 days and described an increase for NIBUT in a Narafilcon A lens but a decrease in a Delefilcon A lens, similar to the one evaluated by us, although dry eye patients were included. The increase in MNIBUT could be explained by the aqueous gradient of the lens [52].

We achieve that the FNIBUT is lower and MNIBUT increase after 30 days. The explanation lies in that the first NIBUT is the time it takes for the tear film to break up after the first blink following the light projection. This measurement provides information only about the initial stability of the tear film. The MNIBUT is the average time it takes for the tear film to break up after multiple blinks following the light projection. This measurement provides information about the overall stability of the tear film. Additionally, the FNIBUT, as a single value measurement, decreased by only 0.32 s (statistically significant but not clinically significant), while MNIBUT increased by 10.12 s (both statistically and clinically significant).

In this work, we verified that the scores on the CLEDQ8 and SPEED tests decreased significantly after one month of CL wearing; therefore, the symptoms were less severe after one month of use. Both questionnaires were previously validated by other authors [29,53], although we found discrepancies in the results. López-de-la-Rosa et al. [26] found no correlation between signs and symptoms, while Chalmers et al. [53] found an improvement in symptoms with the CLDEQ8 questionnaire after two weeks of use, similar to our results. We have to highlight that patients with previous one-day replacement contact lenses were more comfortable with the Lehfilcon A contact lens rather than the habitual one-day.

This research adds novelty to the field by evaluating the effectiveness of Lehfilcon A water gradient silicone hydrogel contact lenses with bacterial and lipid resistance technology in managing dry eye symptoms. This is a prospective, longitudinal, single-centre, self-controlled study, which is unique in its design and provides a more comprehensive understanding of the impact of these contact lenses on ocular surface health. The study uses non-invasive pre-lens tear film analysis with the Integrated Clinical Platform (ICP) Ocular Surface Analyzer (OSA) and Meibomian gland evaluation with the infrared meibographer Cobra^®^ HD, which is a novel and innovative approach to evaluating the effectiveness of contact lenses in managing dry eye disease. The results of the study demonstrate that Lehfilcon A contact lenses significantly reduce lipid and aqueous tear film volume, increase pre-lens non-invasive break-up time, and improve subjective dry eye symptoms.

Regarding the limitations, first, this research failed to compare several water gradient materials or a control group with well-known CL material. Second, the design of the manuscript is based on a refit of contact lenses, which could introduce significant bias. Third, future studies should evaluate both with and without lenses at baseline and after the wearer period. Finally, inflammation biomarkers or cytology impression analysis were not included due to the unavailability of technology. Third, the sample size in the study is limited, with only a small number of participants being evaluated. The power calculation of our study was set at 80%. Future research should include the relationship between the ocular surface and the changes produced by an SH-SCL with objective, automated and non-invasive methods, as well as establish protocols that help to select the suitable CL for each case. In addition, future studies could divide subjects into the dry eye and non-dry-eye groups. Furthermore, a control group with other treatments should be added in future research.

## 5. Conclusions

This study provides valuable insights into the impact of Lehfilcon A water gradient silicone hydrogel contact lenses with bacterial and lipid resistance technology in managing dry eye symptoms. The results demonstrate that these contact lenses have a positive effect on reducing lipid and aqueous tear film volume, increasing pre-lens non-invasive break-up time, and improving subjective dry eye symptoms.

## Figures and Tables

**Figure 1 diagnostics-13-01258-f001:**
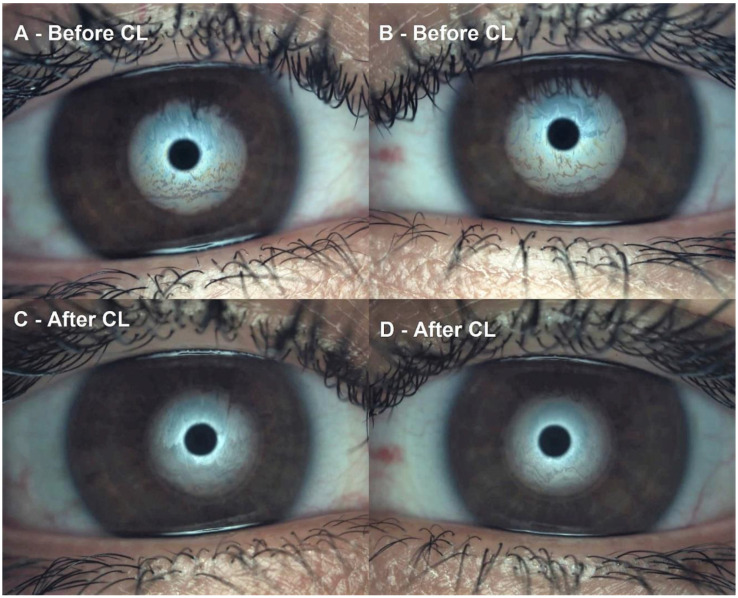
Lipid layer thickness interferometry decreased from grades 5 and 5 to grades 1 and 1: (**A**) Right eye grade 5 Guillon pattern before contact lens, (**B**): Left eye grade 5 Guillon pattern before contact lens. (**C**): Right eye grade 1 Guillon pattern after contact lens and (**D**): Left eye grade 1 Guillon pattern after contact lens. This figure was a singular participant example (subject 12, both eyes).

**Figure 2 diagnostics-13-01258-f002:**
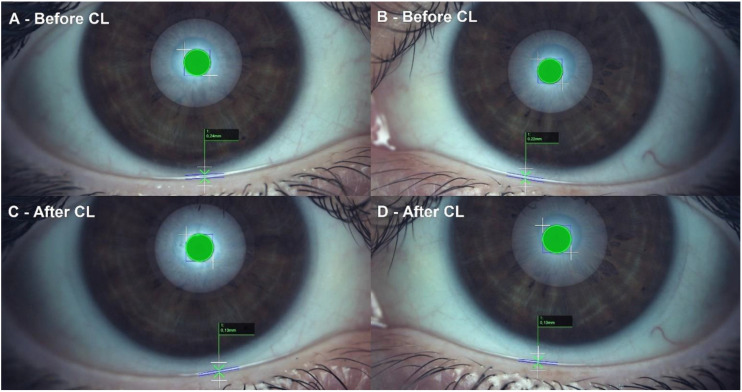
Tear meniscus height decreased from baseline to one-month follow-up. (**A**): Right eye baseline, 0.24 mm, (**B**): Left eye baseline, 0.22 mm, (**C**): Right eye after one month, 0.13 mm and (**D**): Left eye one month, 0.13 mm. This figure was a singular participant example (subject 5, both eyes).

**Figure 3 diagnostics-13-01258-f003:**
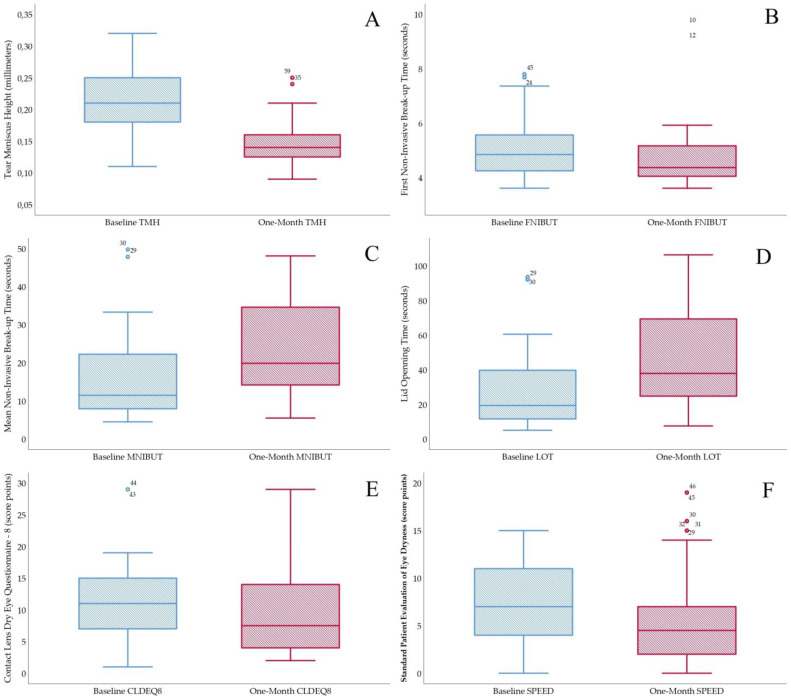
Box & Plot longitudinal changes before and one month after contact lens wear. (**A**): Tear meniscus height, (**B**): First NIBUT, (**C**): Mean NIBUT, (**D**): LOT, (**E**): CLDEQ8 and (**F**): SPEED.

**Table 1 diagnostics-13-01258-t001:** Descriptive analysis of the sample.

Variable	Value
*Gender (%)*	
Male	7 (22.6)
Female	24 (77.4)
*Nationality (%)*	
Italian	21 (67.75)
Spanish	4 (12.90)
Mexican	2 (6.46)
Slovak	1 (3.22)
Polish	1 (3.22)
Germany	1 (3.22)
Austrian	1 (3.22)
Age (years)	22.23 ± 1.39 (19 to 25)
Sphere (Diopters)	−2.64 ± 1.15 (−5.50 to −0.50)
Cylinder (Diopters)	−0.44 ± 0.37 (−1.50 to 0.00)
Axis (Degrees, °)	111.44 ± 70.08 (5.00 to 180.00)
Visual Acuity (Log MAR)	−0.03 ± 0.05 (−0.10 to 0,10)
Visual Acuity (Decimal)	1.07 ± 0.10 (0.80 to 1.20)
Flat Corneal Meridian (mm)	7.87 ± 0.31 (7.40 to 8.74)
Steep Corneal Meridian (mm)	7.73 ± 0.29 (7.25 to 8.61)
Mean Corneal Meridian (mm)	7.80 ± 0.30 (7.37 to 8.67)
Contact Lens Power (Diopters)	−2.56 ± 1.12 (−5.00 to −0.75)
Superior Eyelid MGD (%)	28.87 ± 15.11 (10.30 to 96.20)
Inferior Eyelid MGD (%)	49.69 ± 17.86 (17.00 to 87.30)

BUT: Break-Up Time, CLDEQ8: Contact Lens Dry Eye Questionnaire, SPEED: Standard Patient Evaluation of Eye Dryness.

**Table 2 diagnostics-13-01258-t002:** Ocular surface longitudinal changes before and after silicone hydrogel wearing.

Variable	Baseline	One-Month	*p*-Value
Conjunctival Redness Classification (Efron Scale)	1.08 ± 0.63(0.00 to 2.00)	1.22 ± 0.64(0.00 to 2.00)	0.14
Lipid Layer Thickness Interferometry (Guillon Pattern)	2.05 ± 1.53(0.00 to 5.00)	0.92 ± 1.09(0.00 to 5.00)	<0.01 *
Tear Meniscus Height (mm)	0.21 ± 0.04(0.11 to 0.32)	0.14 ± 0.03(0.09 to 0.25)	<0.01 *
First NIBUT (s)	5.03 ± 1.04(3.60 to 7.80)	4.71 ± 1.10(3.60 to 9.56)	0.09
Mean NIBUT (s)	15.19 ± 9.54(4.50 to 49.76)	25.31 ± 15.81(5.50 to 91.14)	<0.01 *
Lid Opening Time (s)	26.36 ± 19.72(5.04 to 93.60)	46.10 ± 27.45(7.52 to 106.40)	<0.01 *
Schirmer Test (mm)	30.21 ± 8.43(6.00 to 35.00)	20.88 ± 12.72(0.00 to 36.00)	<0.01 *
CLDEQ8 (Score Points)	11.32 ± 5.56(1.00 to 29.00)	10.53 ± 8.23(2.00 to 29.00)	0.49
SPEED Test (Score Points)	7.39 ± 4.39(0.00 to 15.00)	5.53 ± 4.83(0.00 to 19.00)	<0.01 *

NIBUT: Non-Invasive Break Up Time, CLDEQ8: Contact Lens Dry Eye Questionnaire, SPEED: Standard Patient Evaluation of Eye Dryness. * Statistically significant within the Wilcoxon test.

## Data Availability

The data presented in this study are available on request from the corresponding author. The data are not publicly available due to their containing information that could compromise the privacy of research participants.

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
