# Peer review of "Evaluation of Celligent® Biomimetic Water Gradient Contact Lens Effects on Ocular Surface and Subjective Symptoms"

_diagnostics, 2023, doi:10.3390/diagnostics13071258_

Round 1

Reviewer 1 Report

The paper “Evaluation of Celligent® Biomimetic Water Gradient Contact Lens Effects on Ocular Surface and Subjective Symptoms”

conducted by Raúl Capote-Puente et al. provides information on the initial and general stability of the tear film, 30 days after wearing contact lenses.

The subject is relevant for research because it brings new elements regarding the effectiveness of Lehfilcon A water gradient silicone hydrogel contact lenses in the management of dry eye symptoms. The results obtained can be considered by clinicians in choosing this type of medical device for chronic use, both in terms of the benefits of this product, or other similar products, as well as in terms of the safety of their long-term use.

The formulated conclusions are in accordance with the obtained results, based on the proposed study objectives.

The references are relevant to the information presented.

The tables and figures are relevant and present the data accurately, being easy to interpret and understand.

Author Response

Reviewer 1

#RV1: The paper “Evaluation of Celligent® Biomimetic Water Gradient Contact Lens Effects on Ocular Surface and Subjective Symptoms” conducted by Raúl Capote-Puente et al. provides information on the initial and general stability of the tear film, 30 days after wearing contact lenses. The subject is relevant for research because it brings new elements regarding the effectiveness of Lehfilcon A water gradient silicone hydrogel contact lenses in the management of dry eye symptoms. The results obtained can be considered by clinicians in choosing this type of medical device for chronic use, both in terms of the benefits of this product, or other similar products, as well as in terms of the safety of their long-term use. The formulated conclusions are in accordance with the obtained results, based on the proposed study objectives. The references are relevant to the information presented. The tables and figures are relevant and present the data accurately, being easy to interpret and understand.

#AU1: Thank you very much for your positive feedback on our paper "Evaluation of Celligent® Biomimetic Water Gradient Contact Lens Effects on Ocular Surface and Subjective Symptoms". We are pleased to hear that you found our research relevant and informative, particularly in regard to the effectiveness of Lehfilcon A water gradient silicone hydrogel contact lenses for managing dry eye symptoms.

We appreciate that you found our formulated conclusions to be in accordance with the obtained results and that our references, tables, and figures were relevant and accurately presented, making it easy to interpret and understand our findings.

We are grateful for your feedback and hope that our research can provide valuable insights for clinicians and patients alike in choosing medical devices for chronic use, including the benefits and safety considerations associated with the use of these products.

Reviewer 2 Report

It is a well written and organized manuscript; and the methods are adequately described. I have some concerns that the authors should address. I would like the author(s) to address these comments raised by the reviewer.

Line 28: Render “ocular stability” as “stability of ocular surface”

Line 37: Elaborate on these “specialized industries”

Lines 72 – 75: Provide information on the timeline of the study.

Line 84: Meibomitis associated with meibomian gland dysfunction (MGD) can cause ocular surface inflammation, and MGD through the generation of free fatty acids can cause ocular surface inflammation.

Line 113: Replace “nonoptometry” as “non-optometry”

Lines 201 – 203: Mechanical irritation of the ocular surface can trigger an innate immune response with the consequential generation/release of pro-inflammatory cytokines (IL-1, TNF-alpha) that facilitate the generation of pro-lymphangiogenic factors that create channels for DCs to migrate to the secondary lymphoid organs.  Additionally, these pro-inflammatory cytokines can activate resident DCs, and these activated, mature DCs utilize the afferent lymphatic vessels to migrate to the regional secondary lymphoid organs to initiate the generation of effector CD4 T cells (Th1 and Th17 cells).

Lines 226 - 227: The authors should check that these cited references relate to and support the claims made, as the cited references appear to be misplaced. 

Lines 251 – 254: The authors should reword this sentence to render it more comprehensible as well as address the minor punctuation errors. Render “Both questionnaires are well validated, [45,52,53], although there is discrepancy in the results.” as “Both questionnaires are well validated, [45,52,53], although there is a discrepancy in the results”

Lines 268 – 269: The authors should reword this sentence to render it more comprehensible as well as address the minor punctuation errors.

Lines 269 – 272: Consider revising this run-on sentence in order to render it comprehensible.

Lines 273 – 274: Provide a power calculation. It is my belief that this limitation might render the conclusion less convincing. Is the sample size adequate for this analysis?

Author Response

Reviewer 2

#RV0: It is a well written and organized manuscript; and the methods are adequately described. I have some concerns that the authors should address. I would like the author(s) to address these comments raised by the reviewer.

#AU0: Thank you for your feedback on our manuscript. We appreciate your positive comments regarding the writing style, organization, and description of our methods.

We are also grateful for your constructive criticism and would be happy to address any concerns you may have.

#RV1: Line 28: Render “ocular stability” as “stability of ocular surface”

#AU1: Correction made.

#RV2: Line 37: Elaborate on these “specialized industries”

#AU2: Thanks for the comment, we have included the following:

“Specialized industries, including those in the field of contact lens manufacturing, are aimed at improving SCL properties that have the greatest impact on safety, comfort or tolerance of use, such as the module, contact angle, coefficient of friction and oxygen transmissibility.[12,13]”

#RV3: Lines 72 – 75: Provide information on the timeline of the study.

#AU3: Thanks for the comment, we have included the timeline of 30 days.

#RV4: Line 84: Meibomitis associated with meibomian gland dysfunction (MGD) can cause ocular surface inflammation, and MGD through the generation of free fatty acids can cause ocular surface inflammation.

#AU4:  We agree with your comment. We have remove the term inflammation from the exclusion criteria.

#RV5: Line 113: Replace “nonoptometry” as “non-optometry”

#AU5: Correction made

#RV6: Lines 201 – 203: Mechanical irritation of the ocular surface can trigger an innate immune response with the consequential generation/release of pro-inflammatory cytokines (IL-1, TNF-alpha) that facilitate the generation of pro-lymphangiogenic factors that create channels for DCs to migrate to the secondary lymphoid organs.  Additionally, these pro-inflammatory cytokines can activate resident DCs, and these activated, mature DCs utilize the afferent lymphatic vessels to migrate to the regional secondary lymphoid organs to initiate the generation of effector CD4 T cells (Th1 and Th17 cells).

#AU6: Thanks for your valuable comment, we have included in the manuscript as:

“The ocular surface is highly sensitive to mechanical irritation, which can induce an innate immune response [1]. This response triggers the production and release of pro-inflammatory cytokines such as IL-1 and TNF-alpha. These cytokines promote the generation of pro-lymphangiogenic factors, which create channels for dendritic cells (DCs) to migrate to secondary lymphoid organs [2]. Furthermore, the pro-inflammatory cytokines can activate resident DCs, which then utilize afferent lymphatic vessels to travel to regional secondary lymphoid organs. This migration of DCs initiates the generation of effector CD4 T cells, including Th1 and Th17 cells, which are critical components of the adaptive immune response [3].

It is important to note that this cascade of events highlights the complex interplay between innate and adaptive immunity in response to ocular surface irritation. By understanding the underlying mechanisms, we can develop novel approaches to prevent and treat ocular surface diseases associated with inflammation and immune dysregulation [1,2].”

#RV7: Lines 226 - 227: The authors should check that these cited references relate to and support the claims made, as the cited references appear to be misplaced.

#AU7: Thanks for the comment, the references were not link to the reference manager program. We have already included the correct references.

“TMH decreases significantly, to keep the CL hydrated, it is necessary to use the tear film reserve of the TMH [4,5].”

#RV8: Lines 251 – 254: The authors should reword this sentence to render it more comprehensible as well as address the minor punctuation errors. Render “Both questionnaires are well validated, [45,52,53], although there is discrepancy in the results.” as “Both questionnaires are well validated, [45,52,53], although there is a discrepancy in the results”

#AU8: Thanks for the comment, we have reworded the sentence, following your recommendation.

#RV9: Lines 268 – 269: The authors should reword this sentence to render it more comprehensible as well as address the minor punctuation errors.

#AU9: Thanks for the comment, we agree with your consideration. We have reworded as:

“Regarding the limitations, first, this research failed to compare several water gradient materials or a control group with well-known CL material.”

#RV10: Lines 269 – 272: Consider revising this run-on sentence in order to render it comprehensible.

#AU10: Thanks for the comment, we agree with your consideration. We have reworded as:

“Second, the design of the manuscript is based on a refit of contact lens, which could introduce significant bias. Third, future studies should evaluate both with and without lens at baseline and after the wearer period. Finally, inflammation biomarkers or cytology impression analysis were not included due to the unavailability of technology.”

#RV11: Lines 273 – 274: Provide a power calculation. It is my belief that this limitation might render the conclusion less convincing. Is the sample size adequate for this analysis?

#AU11: Thanks for the comment, the sample size was calculated with the Granmo Calculator:

Sample size was estimated using the GRANMO® calculator (Institut Municipal d’Investigació Mèdica, Barcelona, ​​Spain. Version 7.12). A two-tailed test was used. Alpha and beta risks are set at 5% and 20%, respectively. The estimated standard deviation (SD) of the difference was set to 0.45 (based on the SD primary variable study by Marx et al.[6]), and the expected minimum NIBUT difference in front of the camera was set to 0.30 s, followed by the loss at the end. The upward adjustment rate has been fixed at 0.10. This resulted in a recommended sample size of 20 subjects.”

In this case, our research achieved the sufficient sample size to the previously calculated.

The power calculation was an anticipated means of 15.10 ± 9.54 seconds for the mean NIBUT in group 1 and 25.31 ± 15.81 seconds for the mean NIBUT of group 2. The enrollment radio was stablished in 1 and type error I was 0.05 and type error II was 0.20. So the power calculation was 80%.

References

  1. Pflugfelder, S.C.; Stern, M.E. The Cornea in Keratoconjunctivitis Sicca. Exp Eye Res 2020, 201, doi:10.1016/J.EXER.2020.108295.
  2. Yamaguchi, T.; Hamrah, P.; Shimazaki, J. Bilateral Alterations in Corneal Nerves, Dendritic Cells, and Tear Cytokine Levels in Ocular Surface Disease. Cornea 2016, 35, S65–S70, doi:10.1097/ICO.0000000000000989.
  3. Perez, V.L.; Stern, M.E.; Pflugfelder, S.C. Inflammatory Basis for Dry Eye Disease Flares. Exp Eye Res 2020, 201, doi:10.1016/j.exer.2020.108294.
  4. Guillon, M.; Maissa, C. Contact Lens Wear Affects Tear Film Evaporation. Eye Contact Lens 2008, 34, 326–330, doi:10.1097/ICL.0B013E31818C5D00.
  5. Ruiz-Alcocer, J.; Monsálvez-Romín, D.; García-Lázaro, S.; Albarrán-Diego, C.; Hernández-Verdejo, J.L.; Madrid-Costa, D. Impact of Contact Lens Material and Design on the Ocular Surface. Clin Exp Optom 2018, 101, 188–192, doi:10.1111/CXO.12622.
  6. Marx, S.; Eckstein, J.; Sickenberger, W. Objective Analysis of Pre-Lens Tear Film Stability of Daily Disposable Contact Lenses Using Ring Mire Projection. Clin Optom (Auckl) 2020, 12, 203–211, doi:10.2147/OPTO.S262353.

Round 2

Reviewer 2 Report

I am satisfied with your response to my questions; however, I have a couple comments that need to be addressed.

Line 210: In line with the cited reference, render "secondary lymphoid organ" as "lymph nodes"

Line 212: In line with the cited reference, render “secondary lymphoid organs" as "lymph nodes"  

Author Response

Reviewer 2 - Round 2

#RV0: I am satisfied with your response to my questions; however, I have a couple comments that need to be addressed.

#AU0: Thank you for your positive feedback on my response to your questions. We appreciate the opportunity to address your comments and concerns.

#RV1: Line 210: In line with the cited reference, render "secondary lymphoid organ" as "lymph nodes"

#AU1: Correction made.

#RV2: Line 212: In line with the cited reference, render “secondary lymphoid organs" as "lymph nodes"

#AU2: Correction made.